# GOLPH3-mTOR Crosstalk and Glycosylation: A Molecular Driver of Cancer Progression

**DOI:** 10.3390/cells14060439

**Published:** 2025-03-14

**Authors:** Anna Frappaolo, Gianluca Zaccagnini, Maria Grazia Giansanti

**Affiliations:** Istituto di Biologia e Patologia Molecolari del CNR, c/o Dipartimento di Biologia e Biotecnologie, Sapienza Università di Roma, 00185 Roma, Italy; anna.frappaolo@cnr.it (A.F.); gianlucazaccagnini@cnr.it (G.Z.)

**Keywords:** GOLPH3, mTOR, Golgi glycosylation, Drosophila

## Abstract

Originally identified in proteomic-based studies of the Golgi, Golgi phosphoprotein 3 (GOLPH3) is a highly conserved protein from yeast to humans. GOLPH3 localizes to the Golgi through the interaction with phosphatidylinositol-4-phosphate and is required for Golgi architecture and vesicular trafficking. Many studies revealed that the overexpression of GOLPH3 is associated with tumor metastasis and a poor prognosis in several cancer types, including breast cancer, glioblastoma multiforme, and colon cancer. The purpose of this review article is to provide the current progress of our understanding of GOLPH3 molecular and cellular functions, which may potentially reveal therapeutic avenues to inhibit its activity. Specifically, recent papers have demonstrated that GOLPH3 protein functions as a cargo adaptor for COP I-coated intra Golgi vesicles and impinges on Golgi glycosylation pathways. In turn, GOLPH3-dependent defects have been associated with malignant phenotypes in cancer cells. Additionally, the oncogenic activity of GOLPH3 has been linked with enhanced signaling downstream of mechanistic target of rapamycin (mTOR) in several cancer types. Consistent with these data, GOLPH3 controls organ growth in Drosophila by associating with mTOR signaling proteins. Finally, compelling evidence demonstrates that GOLPH3 is essential for cytokinesis, a process required for the maintenance of genomic stability.

## 1. Introduction: GOLPH3, a PI(4)P-Binding Oncoprotein

Golgi phosphoprotein 3 (GOLPH3) was originally identified in proteomic-based studies of isolated Golgi fractions and characterized as a highly phosphorylated protein that dynamically localizes to the cytoplasmic face of the *trans*-Golgi, to the *trans*-Golgi network (TGN) and to plasma membranes [1,2,3]. Further analyses showed that GOLPH3 is a highly conserved protein from yeast to human, which localizes to *trans*-Golgi through its interaction with Phosphatidylinositol 4-phosphate (PI(4)P; [4,5,6]). At the Golgi, the GOLPH3 function is required for the maintenance of the Golgi structure and vesicle trafficking [4,6]. Multiple lines of evidence indicate that GOLPH3 is an oncogene in different types of human solid tumors [7]. Human *GOLPH3* was originally identified as a new oncogene through a gene discovery approach that combined integrative genomics with clinicopathological and functional analyses [7]. Scott and coauthors identified a recurrent 5p13 amplification site in several solid tumors, including melanoma, breast, colorectal cancer, and non-small-cell lung cancer [7]. The 5p13 amplification was shown to correlate with the expression of two genes in human lung cancer specimens (namely, *GOLPH3* and *SUB1*). However, gene silencing coupled with the cDNA overexpression of both genes pinpointed *GOLPH3* alone as the gene targeted for activation in cancers with 5p13 amplification. Consistent with these results, *GOLPH3* overexpression promoted the malignant transformation of both primary non-transformed mouse and human cells and enhanced mouse xenograft tumor growth *in vivo* [7]. To gain insight into the biological role of GOLPH3, Scott and coworkers performed a yeast-two hybrid screen aimed at identifying GOLPH3-molecular partners. This analysis revealed the GOLPH3 interaction with the VPS35 subunit of the retromer protein-recycling complex, which has been linked to mechanistic target of rapamycin (mTOR) signaling in budding yeast [7,8]. The evolutionarily conserved target of rapamycin (TOR) is a nutrient-sensing protein kinase that regulates cellular metabolism, cell growth, and division [9,10]. In most organisms, the TOR kinase represents the central catalytic subunit of two distinct functional multi-subunit complexes known as mechanistic target of rapamycin complex 1 (mTORC1) and mTORC2 with different cellular functions and substrates [9,10]. Importantly, in non-small-cell lung cancer specimens, the 5p13 amplification correlated with an increased mTOR expression and phosphorylation of the TOR substrate p70 S6 kinase, supporting the hypothesis that GOLPH3 might control mTOR activity in mammalian cells [7]. Consistent with a possible link between GOLPH3 and mTOR signaling, the overexpression of GOLPH3 caused the enhanced activation of both mTORC1- and mTORC2-specific substrates [7]. Furthermore, when transplanted in immunodeficient mice, GOLPH3-overexpressing human tumor cells developed tumors faster than the control and displayed an increased sensitivity to rapamycin, a potent mTORC1 inhibitor, indicating that the oncogenic activity of GOLPH3 is mediated through mTOR signaling [7]. Since its validation as an oncogene, the overexpression of *GOLPH3* mRNA and protein has been reported in a variety of solid tumors [11,12] and has been correlated with clinical progression and a poor prognosis in breast cancer, lung cancer, prostate cancer, esophageal squamous cell carcinoma (ESCC), oral tongue cancer, pancreatic ductal adenocarcinoma, and glioblastoma multiforme (GBM) [13,14,15,16,17,18,19,20,21,22]. Moreover, gain- and loss-of-function analyses associated GOLPH3 expression with cancer phenotypes in human breast cancer and GBM cancer cells *in vitro* and *in vivo* [14,21].

Although the role of GOLPH3 as a cancer driver in human solid cancers is well-established, the precise link between GOLPH3 and tumorigenesis is still unclear. In this review, we summarize the recent progress in dissecting the cellular and molecular functions of GOLPH3 that may lead to new therapeutic interventions for cancer.

## 2. Role of GOLPH3 Family Proteins in Golgi Structure Maintenance and Glycosylation

The Golgi apparatus has a central role in the endomembrane-trafficking pathways as well as in protein and lipid glycosylation [23,24]. Defects of the Golgi architecture and trafficking are commonly observed in cancer cells and have been correlated with cancer development and progression [25]. Compelling evidence has shown that PI(4)P and its interacting protein GOLPH3 play an essential role in maintaining Golgi structure and function [4,6]. Dippold and coauthors reported that GOLPH3 interacts with both PI(4)P and unconventional Myosin 18A, which, in turn, binds F-actin [4]. They proposed a model whereby the PI(4)P-GOLPH3/Myo18A/F actin module applies a tensile force to the Golgi which stretches the Golgi membranes and controls vesicle budding for forward trafficking [4].

A link between PI(4)P-GOLPH3-dependent vesicular trafficking and cancer is through the Phosphoinositide Transfer Protein Cytoplasmic 1 (PITPNC1; [26]) protein. *PITPNC1* is an oncogene that is amplified or overexpressed in several metastatic breast, melanoma, and colon cancers [26]. Halberg and coworkers demonstrated that PITPNC1 interacts with Golgi-resident PI(4)P and localizes Rab1B which facilitates the Golgi recruitment of GOLPH3, Golgi extension, and enhanced vesicular release. The PITPNC-Rab1-GOLPH3 network drives the malignant secretion of pro-invasive and pro-angiogenic mediators, which, in turn, leads to cancer phenotypes, metastasis, and angiogenesis.

PI(4)P-GOLPH3 also affects glycosylation, an essential cellular process which has a great impact on cancer signaling, tumor progression, and metastasis [27]. Much evidence in human cells and model organisms indicates that the oncogenic properties of GOLPH3 are strictly linked to its role in regulating the spatial distribution of glycosyltransferases within the Golgi cisternae [28]. Golgi glycosyltransferases are all type II membrane proteins with a short N-terminal cytosolic region (less than 25 amino acids), followed by a single membrane-spanning segment and a luminal catalytic domain [29]. The retention of Golgi resident proteins requires coatomer-coated (COPI) vesicle-mediated Golgi vesicle trafficking. However, the cytoplasmically exposed portion of Golgi glycosyltransferases lacks the canonical COPI-binding motifs [30]. Two studies initially implicated the GOLPH3 yeast ortholog Vps74p in the retention of glycosyltransferases [31,32]. Vps74p can simultaneously interact with a pentameric motif (F/L)(L/I/V)XX(R/K) in the cytoplasmic tails of numerous mannosyltransferases as well as with multiple subunits of the COPI coat [31,32]. This interaction is essential for the Golgi localization of these enzymes, leading to the proposal that Vps74p functions as a glycosyltransferase-sorting receptor for the COPI coat [31,32]. An analysis of the molecular structure of GOLPH3 family proteins revealed that both GOLPH3 and Vps74p proteins form homo-oligomers and that oligomeric Vps74p interacts with PI(4)P and with the pentameric sequence motif at the cytosolic exposed tail of glycosyltransferases [5,32]. Vps74p was shown to recognize and bind the consensus amino acid sequence (F/L)(L/I/V)XX(R/K) in the cytoplasmic tails (CT) of yeast mannosyltransferases [5,32]. The expression of human GOLPH3 rescued the phenotypic defects in *vps74*-null yeast cells [31], indicating that GOLPH3 and Vps74p share similar functions at the Golgi. In addition, Tu and coworkers identified an evolutionarily conserved cluster of arginine residues at the N-terminal end of GOLPH3 proteins that serves to mediate coatomer binding [33]. The loss of the Vps74p-coatomer interaction impairs the Golgi localization of glycosyltransferases without compromising Vps74p recruitment to Golgi membranes, which, instead, depends on the oligomerization status and on PI(4)P binding [33]. Much evidence indicates that, similar to Vps74p, GOLPH3 protein can direct specific glycosyltransferases into COPI vesicles [28,34,35,36]. The (F/L)(L/I/V)XX(R/K) sequence recognized by Vps74p is also present in the cytoplasmic tail of core 2 N-acetylglucosaminyltransferase 1 (C2GnT1) [34]. C2GnT is a key enzyme for the synthesis of core 2-associated sialyl Lewis x (C2-O-sLex), a ligand involved in selectin-mediated leukocyte trafficking and cancer metastasis [34]. Ali and coworkers employed C2GnT1 CT as a bait in yeast two-hybrid experiments to identify GOLPH3 as a molecular partner of C2GnT1. They further showed that C2GnT1 and GOLPH3 co-localized at the Golgi and that GOLPH3 knockdown relocated C2GnT1 in the endoplasmic reticulum, impairing the synthesis of C2-O-sLex associated with P-selectin glycoprotein ligand-1 [34]. A different approach based on a T7 phage was used to identify the interaction between GOLPH3 and Protein O-Linked Mannose β-1,2-N-Acetlyglucosaminyltransferase 1 (POMGnT1), a glycosyltransferase involved in the O-mannosylation of α-dystroglycan. α-Dystroglycan is a key molecular component of the dystrophin glycoprotein complex which mediates the interactions with several proteins of the extracellular matrix including laminin [37]. Importantly, alterations in the O-mannosylation of α-dystroglycan reduces ligand binding, leading to various forms of congenital muscular dystrophies. The loss of the GOLPH3/POMGnT1 interaction disrupts POMGnT1 localization to the Golgi membranes, causing the defective glycosylation of α-dystroglycan [35]. Further investigation will clarify whether the loss of GOLPH3 might be involved in the development of the muscle–eye–brain disease.

The involvement of GOLPH3 in glycosylation was also shown by studies in *Drosophila melanogaster* reporting that the fly homolog regulates the biosynthesis of heparan sulfate glycosaminoglycan (GAG) chains by modulating the retrograde trafficking of Golgi exostosin (EXT) glycosyltransferases [38]. Exostosins catalyze the repetitive addition of β1-4-linked glucuronic acid (GlcA) and α1-4-linked N-acetylglucosamine (GlcNAc) to generate GAG chains [38]. Importantly, in humans, the tumor suppressors EXT1 and EXT2 have been associated with hereditary multiple osteochondroma (MO), an autosomal dominant skeletal disease in which patients develop multiple benign bone tumors [39,40]. Chang and coworkers demonstrated that Drosophila GOLPH3 physically interacts with the fly orthologs of human EXT1 and EXT2. Both the loss or overexpression of GOLPH3 affect the steady-state distribution of EX1 and EX2 enzymes within the Golgi cisternae, resulting in the incomplete biosynthesis of heparan sulfate proteoglycans (HSPGs) and a reduction in Hedgehog signaling [38]. Consistent with the results in Drosophila, the Golgi retention of EXTs requires human GOLPH3 in osteosarcoma cells (U2OS and MG63), chondrosarcoma cells (SW1353), a chondrocyte cell line (CHON-002), and rhabdomyosarcoma (RD) cells [38]. We recently provided evidence that Drosophila GOLPH3 interacts with glycosyltransferase enzymes that control multiple glycosylation pathways. By using an approach based on affinity purification coupled with mass spectrometry (AP-MS), we have characterized the protein–protein interaction network (interactome) of Drosophila GOLPH3 [41]. Consistent with the results from Chang and coworkers, our data indicated that GOLPH3 binds COPI subunits as well as the exostosin Brother of tout-velu (Botv)/EXTL3, a glucuronyl-galactosyl-proteoglycan 4- alpha-N-acetylglucosaminyltransferase required for HSPG synthesis [38]. The additional molecular partners of Drosophila GOLPH3 identified in our study suggest the involvement of GOLPH3 in the synthesis of mucin-type O-glycans and glycosylphosphatidylinositol (GPI) anchor processing [41].

Two recent studies in human cells support a role for GOLPH3 in glycolipid synthesis [42,43]. Glycosphingolipids are the most represented class of glycolipids in vertebrates and critical components of plasma membranes where they regulate cell signaling and cell–cell communication [44]. Rizzo and coworkers demonstrated that GOLPH3 specifically bound and retained in the Golgi, a selected group of key enzymes that operate at the branchpoint among glycosphingolipid synthetic pathways [42]. One such enzyme is lactosylceramide synthase (LCS), which converts glucosylceramide to lactosylceramide (LacCer) in the *trans*-Golgi/TGN. The depletion of GOLPH3 resulted in increased LCS trafficking to lysosomes, whereas the overexpression of GOLPH3 enhances LCS retention in the Golgi and affects its transfer to lysosomes (Figure 1a).

Importantly, modulating the levels of GOLPH3 had similar effects on the cellular localization of other GOLPH3 client enzymes. In contrast, other *trans*-Golgi-located enzymes that do not bind GOLPH3, such as sphingomyelin synthase 1 and glucosylceramide synthase, were insensitive to manipulations of GOLPH3 levels. The GOLPH3-induced changes on its client enzymes impacted the glycosphingolipid metabolism both in primary human fibroblasts (PHFs) and in HeLa cells, leading to the corresponding changes in the abundance of LacCer and other complex glycosphingolipids. Based on these findings, the authors wondered whether the oncogenic potential of GOLPH3 could be related to the effects on glycosphingolipid synthesis. Importantly, they found that LCS overexpression mimicked the effects of GOLPH3 overexpression, leading to enhanced Akt/mTORC1 signaling as well as in assays of cell proliferation. Finally, the levels of LCS protein and GOLPH3 proteins were significantly correlated in non-small-cell lung cancer samples from human patients [42]. Overall, the results from this work suggest that GOLPH3 may promote mitogenic signaling and cell proliferation at least in part by enhancing the synthesis of glycosphingolipids ([42], Figure 1a).

By using biochemical and fluorescence microscopy approaches, Ruggiero and coworkers explored the involvement of GOLPH3 in the metabolism of complex sialyl-glycolipids in GBM (T98G) and breast cancer (MCF7) cell lines [43]. GOLPH3 knockdown caused a downregulation of GD1a, accompanied with a concomitant GM1 upregulation at the cell surface, indicating a role for GOLPH3 in the surface expression of complex sialyl-glycolipids. They demonstrated that GOLPH3 physically interacts with both ST3Gal-II (β-Galactoside α-2,3-Sialyltransferase 2, GD1a synthase) and β3GalT-IV (β-1,3-galactosyltransferase 4, GM1 synthase) in mammalian cultured cells. Surprisingly, the depletion of GOLPH3 does not affect the localization of both enzymes to the Golgi but impairs the formation of the ST3Gal-II/β3GalT-IV protein complex, causing a change in the glycolipid expression pattern. These findings suggest a novel GOLPH3-mediated control mechanism of cellular glycosylation through the organization of multienzymatic glycosyltransferase complexes in the Golgi [43].

Work from the group of S. Munro suggested that GOLPH3 might play a more general role in the retention of glycosylation enzymes [45]. Welch and coworkers applied an approach based on two orthogonal, unbiased proteomic methods to identify clients for GOLPH3 and GOLPH3L, a paralog expressed at low levels in most tissues [45]. Their analysis showed that GOLPH3 and GOLPH3L bind a large array of Golgi resident enzymes that act in several glycosylation pathways including mucin-type O-linked mucin-type glycosylation, N-linked glycosylation, O-mannosylation, and proteoglycan synthesis. By combining *in vitro* binding studies, a bioinformatic analysis, and *in vivo* Golgi retention assays, they demonstrate that GOLPH3 and GOLPH3L bind the cytoplasmic tails of a wide range of Golgi enzymes through a membrane-proximal cluster of positively charged residues. Moreover, the deletion of GOLPH3 and GOLPH3L leads to broad-spectrum defects in glycosylation, as proven by probing with a panel of fluorescently labeled lectins. Based on their findings, the researchers propose that GOLPH3 and GOLPH3L function as major COPI adaptors that could influence most, if not all, glycosylation pathways [45].

Overall, these data strongly indicate a link between the functional role of GOLPH3 in glycosylation and its oncogenicity. It is well-supported that altered glycosylation is a hallmark of cancer cells, which plays a pivotal role in tumor phenotypes and cancer progression towards invasiveness [46,47]. Changes in N-glycans, O-glycans, and glycosphingolipids are frequently enriched on the tumor cell surface and have been characterized as cancer biomarkers that can improve early diagnosis and personalized treatments [48,49]. For example, sialic-acid-containing glycan antigens such as sialyl Thomsen-nouvelle antigen (sialyl Tn) and sialyl Lewis (sLe) on serum proteins are frequently used as diagnostic markers and correlate with metastasis and poor prognosis [50]. Additionally, the upregulation of glycoprotein sialylation has been associated with several types of malignancies including breast, ovary, and colorectal cancers [51,52,53,54,55]. A role for human GOLPH3 in N-glycan sialylation is well-supported [36,56,57]. Isaji and coworkers showed that GOLPH3 knockdown in HeLa cells impairs integrin-dependent cell migration and decreases the sialylation of N-glycans. Importantly, N-glycans of integrin β1 is one of the relevant targets for the sialylation and depletion of GOLPH3 results in integrin β1 hyposialylation [56]. Furthermore, GOLPH3 interacts with α2,6-sialyltransferase-I (ST6GAL1) and the overexpression of ST6GAL1 suppresses integrin-dependent cell migration defects in GOLPH3 KD cells [56]. Consistent with a role in sialylation, GOLPH3 controls the incorporation of Core 2N-acetylglucosaminyltransferase (C2GnT) and ST6GAL1 into COPI vesicles as well as the subcellular localization of these enzymes [36]. In contrast, galactosyltransferase does not interact with GOLPH3 and is not dependent on GOLPH3 for proper localization [36]. GOLPH3 has a specific affinity for PI(4)P, which is highly enriched in the TGN, and is produced mainly by PI(4) kinases [PI(4)K] [4]. In MDA-MB-231 breast cancer cells, the biosynthesis of PI(4)P at the TGN requires PI(4)Kα, whereas PI(4)KIIIβ is mainly localized at the cis-Golgi [58,59]. Importantly, some studies have reported the biochemical association between PI(4)Kα and integrin β1, suggesting a possible involvement of integrin in the biosynthesis of PI(4)P [60,61,62]. Moreover, recent papers characterized a regulatory mechanism underlying sialylation in MDA-MB-231 breast cancer cells, mediated by the regulatory interplay between integrin α3β1-PI(4)Kα and GOLPH3-α3-sialyltransferase [57,63,64]. One study explored the contribution of PI(4)Kα to sialylation in MDA-MB-231 breast cancer cells, showing that the depletion of PI(4)Kα impairs the synthesis and localization of PI(4)P in the TGN, decreases N-glycan sialylation on the cell surface, and affects Akt phosphorylation, and integrin α3-mediated cell migration [63]. Furthermore, both integrin α3β1 and PI(4)Kα co-localize at the TGN, and PI(4)Kα specifically interacts with integrin α3 and not with integrin α5. In agreement with the functional role of the PI(4)Kα-integrin α3 complex in sialylation, the loss of integrin α3 significantly disrupts the N-sialylation of membrane glycoproteins, including the epidermal growth factor receptor [63]. A more recent article demonstrated that Focal adhesion kinase (FAK), a downstream effector of integrin β1, has a crucial role in the PI(4)KIIα-GOLPH3-sialyltransferase axis [64]. Knockout of FAK in HeLa cells affects PI(4)KIIα expression levels via the proteasomal degradation pathway. This, in turn, impairs the formation of the GOLPH3-sialyltransferases, resulting in decreased sialylation (Figure 1b).

## 3. GOLPH3 Enhances Signaling Through mTOR

Compelling evidence indicates that GOLPH3 and its interaction with PI(4)P modulate the mTOR signaling pathway, which controls cellular growth, metabolism, and survival [7]. An analysis of many solid tumors has revealed that the upregulation of GOLPH3 is associated with the hyperactivation of growth factor-induced mTOR signaling [7,12]. Importantly, more than 70% of cancers display an aberrant hyperactivation of mTOR, a promising target for therapies against human malignancies [9]. The mTOR kinase is associated with two functional multiprotein complexes named mTORC1 and mTORC2, which differ in their subunit composition and the response to rapamycin [65]. mTORC1 and mTORC2 share the mTOR kinase and the mammalian lethal with SEC13 protein 8, Lst8 in *D. melanogaster* (mLST8) subunit, while specific components are unique to each complex [65]. mTORC1 contains regulatory-associated protein of mTOR (Raptor), DEP-domain-containing mTOR-interacting protein (DEPTOR), and the 40 kDa proline-rich Akt substrate (PRAS40). mTORC2 consists of rapamycin-insensitive companion of mTOR (Rictor), Protor/Proline-rich protein 5 (PRR5), DEPTOR, and mammalian stress- activated protein kinase-interacting protein 1 (mSIN1) [65]. The small GTPase protein Ras homolog enriched in brain (Rheb), in its GTP-bound state, acts as an activator of mTORC1, whereas the heterotrimeric tuberous sclerosis complex (TSC) functions as a GTPase-activating protein (GAP) and a negative regulator of Rheb [66,67,68,69,70]. Much evidence indicates that GOLPH3 influences tumorigenesis by enhancing the activities of both mTORC1 and mTORC2, resulting in the elevated phosphorylation of their respective substrates [7,12].

Based on the cellular functions of GOLPH3, several hypotheses have been formulated for the molecular mechanisms that link its oncogenicity to mTOR signaling. One initial hypothesis has involved the interaction with VPS35, a component of the retromer complex that regulates endosome-to-Golgi trafficking, raising the possibility that GOLPH3 might affect mTOR complexes by promoting the endocytosis and recycling of key transmembrane receptors [7,12]. The upregulation of GOLPH3 might also influence mTOR signaling through its effects on glycosylation, which influences the endocytosis and recycling of cancer-relevant glycoproteins that act upstream of the mTOR complex, leading to prolonged growth factor signaling [63,71]. In the context of glycosylation, it has been proposed that GOLPH3 controls glycosphingolipid synthesis and plasma membrane composition, which, in turn, enhances Akt/mTORC1 signaling and promotes cancer cell growth [42] (Figure 1a).

According to the most accredited model, mTORC1 localizes to the surface of lysosomes, where it is activated by GTP-bound Rheb [72,73]. However, the lysosomal localization of Rheb is controversial and several mTORC1 components as well as Rheb protein have been found associated with the Golgi membranes [74,75,76,77,78,79]. Moreover, a recent article has reported that Rheb localized on the Golgi membranes activates mTORC1 at a novel Golgi-lysosome inter-organelle contact site [76]. Overall, these data have led to the proposal that the Golgi apparatus may function as a hub for mTORC1 activation [74,75,76,77,78,79].

Consistent with this hypothesis, we have recently demonstrated that PI(4)P-GOLPH3 controls organ growth in *Drosophila melanogaster* by directly affecting the Rheb-mTORC1 axis together with Translationally controlled tumor protein (Tctp) [80,81]. Previous research studies in *D. melanogaster* have shown that Tctp and 14-3-3 proteins are required for controlling organ growth by regulating Rheb GTPase activity [82,83,84]. We have shown that GOLPH3 physically interacts with Tctp and 14-3-3ζ proteins [41,80]. The wing- and eye-specific depletion of dGOLPH3 results in a significant reduction in the organ size mimicking the effects of mutations in *Tctp* [80]. This phenotype is partially rescued by overexpressing Tctp, 14-3-3ζ, or Rheb proteins. The Golgi localization of Rheb in Drosophila cells requires PI(4)P-GOLPH3, and a mutant version of GOLPH3 that is unable to bind PI(4)P fails to bind Rheb protein, suggesting that GOLPH3 associates with Rheb at the Golgi apparatus. Additionally, knockdown of GOLPH3 reduces the efficiency of Tctp-Rheb complex formation, indicating a role for GOLPH3 in promoting the Tctp-Rheb association. We have proposed a model whereby GOLPH3 is required to recruit Rheb to the Golgi apparatus and to support the Tctp-Rheb-mediated activation of mTORC1. In agreement with a role in mTORC1 activation, the depletion of GOLPH3 reduces the levels of phosphorylated S6K, a downstream target of mTORC1 (Figure 2).

Furthermore, our findings in *D. melanogaster* provided evidence for a physical association of GOLPH3 with Lst8, suggesting a potential link with mTORC2 activity [41,80]. Importantly, although mLST8/Lst8 has been identified as a shared component of mTORC1 and mTORC2, work in Drosophila and mammalian cells has indicated that this protein is required for mTORC2 function but appears not to affect mTORC1 function [85,86]. In this context, a recent paper investigated the role of mLST8 in a panel of normal and cancer cells showing that mLST8 loss, or even a single pair of mutations affecting mTOR binding, completely impaired mTOR association with mTORC2 cofactors RICTOR and SIN1 without affecting mTORC1. This study further showed a direct interaction between mLST8 and the mTORC2 cofactor SIN1. leading to the proposal that mLST8 acts as a scaffold protein for the assembly and activity of mTORC2 [87]. Because, in most solid tumors, the upregulation of GOLPH3 is associated with the enhanced activation of both mTORC1 and mTORC2, it will be important to further dissect the potential interplay of GOLPH3 with mLST8 in mTORC2 signaling in a broad set of cancer contexts.

## 4. GOLPH3 Plays an Essential Function During Cytokinesis

Another possible route through which GOLPH3 function might be linked to tumorigenesis is via the process of cytokinesis. We have provided compelling evidence that GOLPH3 plays an essential role during cytokinesis, thus contributing to the maintenance of genome stability [6,12,88]. Our studies in Drosophila revealed that GOLPH3 protein localizes to the cleavage furrow of dividing spermatocytes and neuroblasts and interacts with proteins of the vesicle trafficking and cytokinesis machineries [6,12,88]. The loss of GOLPH3 causes defects of male meiotic cytokinesis, resulting in multinucleate spermatids [6]. In addition, larval brains depleted of GOLPH3 display polyploid neuroblasts, indicating that GOLPH3 is also required for cytokinesis in somatic cells.

In animal cell cytokinesis, the interaction of non-muscle myosin II (NMII) with F-actin in the contractile ring generates the dominant force that pinches the mother cell into two daughters at the end of cell division [89]. At the heart of cytokinesis, the centralspindlin complex plays critical functions for central spindle assembly and Rho-GTP-dependent contractile ring assembly [89]. We have shown that GOLPH3 binds the centralspindlin complex protein Pavarotti (Pav/MKLP1) and cooperates with myosin II for the stabilization of centralspindlin at the cleavage site and contractile ring dynamics (Figure 3) [41,90].

The GOLPH3 function in cytokinesis is intimately connected to its ability to bind PI(4)P, suggesting that it acts as a key molecule in coupling PI(4)P signaling with membrane remodeling and actomyosin ring dynamics. Importantly, phospho-mTORSer2481, the Ser2481-autophosphorylated form of mTOR, has been found enriched in the central spindle midzone of dividing HeLa cells during the telophase, suggesting a possible involvement of active mTOR in cytokinesis [91]. These findings suggest the GOLPH3-mTOR interplay might also regulate cytokinesis.

The involvement of GOLPH3 in cytokinesis is of great interest in cancer biology as it is well-established that genetically unstable tetraploid cells generated from cytokinesis failures promote tumorigenesis, chromosomal instability (CIN), and, consequently, aneuploidy, resistance to anti-cancer therapy, and cancer evolution [92]. On the other hand, recent work suggests that numerical chromosomal abnormalities and whole-genome doubling represent a common feature of cancer cells which confers a high instability and unique vulnerabilities that are being exploited therapeutically [93,94]. Indeed, a growing body of evidence indicates that inducing cytokinesis failures could provide a potential anti-cancer therapeutic means for several types of cancers characterized by active cell proliferation and polyploidy [92,95,96]. The rationale behind this strategy is to selectively kill polyploid cancer cells that are much more sensitive to cytokinesis failures. This approach has proven to be effective in GBM cells, where treatments with cytokinesis inhibitors cause cytokinesis failures and either growth arrest and apoptosis or senescence, with the latter response observed in highly polyploid cells [97,98]. Importantly, recent work in Drosophila GBM models led to the identification of a conserved cytokinesis pathway which regulates neoplastic proliferation by driving cytokinesis in GBM cells [99].

## 5. Conclusions

An increasing number of studies has provided strong support for the notion that GOLPH3 overexpression is a cancer driver and can be used as a positive biomarker for tumor progression and poor prognosis [12,100,101,102]. Thus, GOLPH3 has been proposed as a potential effective therapeutic target in several solid tumors [102]. Importantly, GOLPH3 overexpression has been reported to confer resistance to the DNA-damaging compounds camptothecin or doxorubicin [103]. Drug resistance is one of the most challenging obstacles in medical oncology, allowing cancer progression and tumor relapse [104]. This problem arises from a wide range of intrinsic and extrinsic cues such as tumor heterogeneity, epigenetic and genetic changes, the immune system, and the surrounding microenvironment [105]. Knockdown of GOLPH3 reduces cell survival after DNA damage, suggesting that GOLPH3 inactivation might be exploited in combination with DNA-damaging compounds as a strategy to enhance the effects of chemotherapy [103].

Thus, dissecting the molecular mechanisms through which GOLPH3 impacts mTOR signaling, cytokinesis, and glycosylation (Appendix A) will contribute to identifying novel therapeutic anticancer strategies. In this context, a growing body of evidence suggests that targeting cytokinesis factors and inducing cytokinesis failures could provide a promising strategy to selectively target highly proliferating and polyploid cancer cells [92,95,96]. Consistently, a number of clinical trials are currently testing the efficacy of drugs impacting tumor cell cytokinesis. Therefore, a comprehensive knowledge of the molecular mechanisms underlying GOLPH3 function and regulation in cytokinesis could lead to the identification of new therapeutic biomarkers and new strategies to treat tumors characterized by GOLPH3 upregulation.

Taking into account the role of GOLPH3 as a COPI adaptor with an essential role in Golgi glycosylation [45], it will be also important to investigate the landscape of the effects of GOLPH3 on glycome profiles in cancer cells. Importantly, not only can glycoconjugates offer new biomarkers for patient stratification, but innovative therapeutic strategies are also currently exploiting the aberrant glycosylation of cancer cells [46].

## Figures and Tables

**Figure 1 cells-14-00439-f001:**
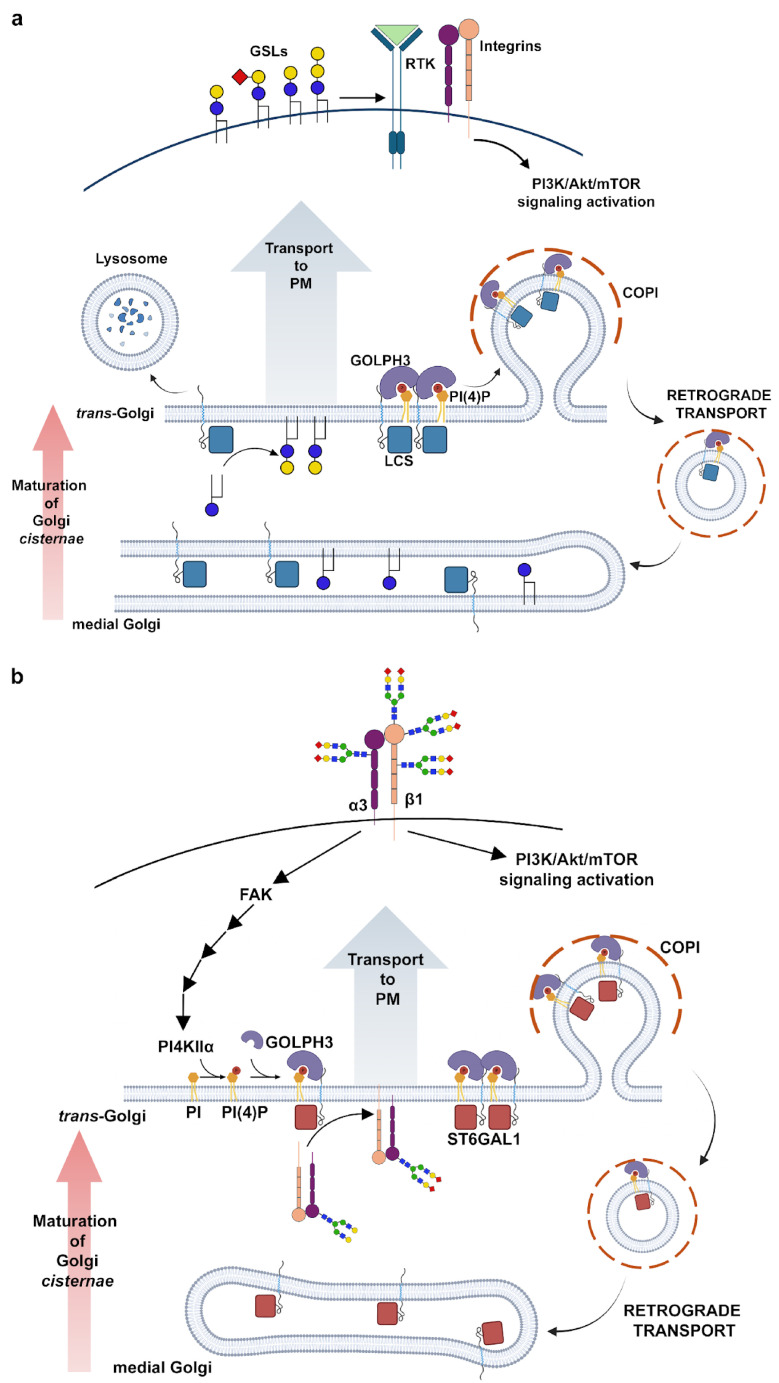
GOLPH3 regulates the spatial distribution of glycosyltransferases within the Golgi cisternae. (**a**) Schematic diagram depicting the role of GOLPH3 in GSL synthesis. GOLPH3 localizes at the *trans*-Golgi through its interaction with locally synthesized PI(4)P. GOLPH3 simultaneously interacts with the COPI coat and the cytoplasmic tails of specific glycosyltransferases, thereby mediating retrograde transport of these enzymes into COPI-coated vesicles. Synthesis of complex GSLs requires action of several Golgi glycosyltransferases. LCS is one such enzyme and converts glucosylceramide to lactosylceramide. GOLPH3 acts as a component of the cisternal maturation mechanism and controls the balance between the sub-Golgi localization and the lysosomal degradation rate for LCS and other specific enzymes. GSLs, synthesized at the Golgi, are transported to the plasma membrane where they activate tyrosine kinase receptors (RTKs) and integrins. By enhancing the rate of enzyme retention in the Golgi, GOLPH3 promotes biosynthesis of GSLs, which, in turn, promotes cell growth through PI3K-Akt-mTOR signaling (blue circle, glucose; yellow circle, galactose; red square, sialic acid). (**b**) Schematic diagram depicting the role of GOLPH3 in integrin sialylation. GOLPH3 accumulation at the *trans*-Golgi requires binding to PI(4)P, that is generated by phosphatidylinositol 4-kinase-IIα [PI(4)KIIα]-mediated phosphorylation of phosphatidylinositol (PI). PI(4)P-GOLPH3 forms a complex with α2,6-sialyltransferase-I (ST6GAL1) which is essential for sialylation of integrins. Proper sialylation of integrins activates the PI3K-Akt-mTOR signaling pathway. The integrin β1 downstream effector FAK stabilizes [PI(4)KIIα], ensuring PI(4)P accumulation at the *trans*-Golgi. In addition, by mediating ST6GAL1 incorporation into COPI-coated vesicles, GOLPH3 controls the correct distribution of sialyltransferases within the Golgi cisternae (red square, sialic acid; blue square, N-acetylglucosamine; green circle, mannose; yellow circle, galactose).

**Figure 2 cells-14-00439-f002:**
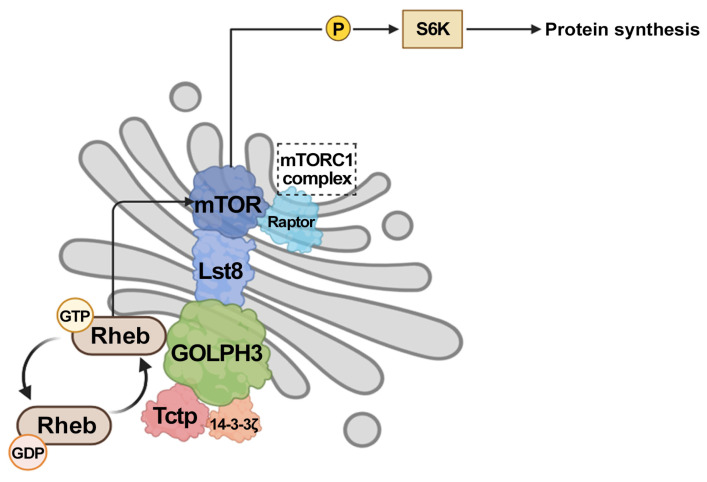
Schematic depicting a model for the role of GOLPH3 in mTORC1 signaling in *Drosophila melanogaster*. mTORC1-mediated phosphorylation and activation of p70 S6 kinase I (S6K) promote protein synthesis and cell growth. GTP-bound Rheb functions as an activator of mTORC1. Tctp controls organ growth by forming a complex with 14-3-3 proteins and acting as a GEF for Rheb. PI(4)P-bound GOLPH3 controls localization of Rheb protein to the Golgi membranes and facilitates Tctp-Rheb association. GOLPH3 also interacts with the Lst8 protein.

**Figure 3 cells-14-00439-f003:**
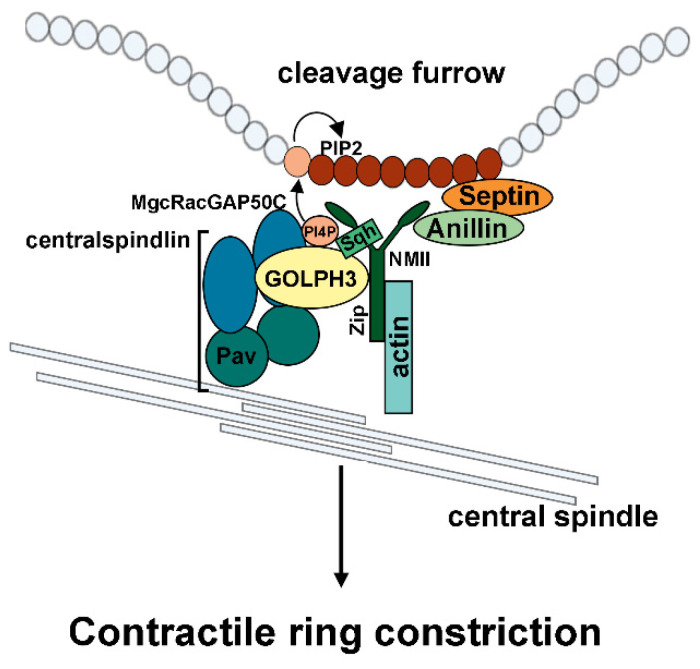
Schematic of Drosophila GOLPH3 involvement in cytokinesis. PI(4)P-GOLPH3 localizes at the cleavage furrow of dividing cells and cooperates with non-muscle myosin II (NMII) proteins [Spaghetti Squash (Sqh) and Zipper (Zip)] to regulate stabilization of centralspindlin and contractile ring structure. PI(4)P [PI4P] accumulates at the cleavage furrow membrane where it is converted to PI(4,5)P2 [PIP2]. Both PI(4)P-GOLPH3 and PI(4,5)P2 stabilize centralspindlin and the contractile ring components at the cleavage furrow.

## Data Availability

No new data were created or analyzed in this study.

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
