# Peer review of "GOLPH3-mTOR Crosstalk and Glycosylation: A Molecular Driver of Cancer Progression"

_cells, 2025, doi:10.3390/cells14060439_

Round 1

Reviewer 1 Report

Comments and Suggestions for Authors

In this review article, Frappaolo et al., summarize the recent progress in dissecting the cellular and molecular functions of Golgi phosphoprotein 3 (GOLPH3) that may lead to new therapeutic interventions for cancer. The manuscript is well written and cover several aspects regarding the functions of the GOLPH3. However I have some suggestions, and hope it may improve the quality of the manuscript. 

-Regarding the impact of GOLPH3 on tumor biology, I suggest the authors describe in more detail the role of the protein, which is known to be located in the Golgi apparatus, in events such as described below.

- GOLPH3 and its interaction with Phosphatidylinositol 4-phosphate (PI4P) is an essential phenomenon for maintaining the structure and function of the Golgi apparatus. This interaction modulates the activation of the mTOR pathway, which is central to regulating cell growth, metabolism, and survival. GOLPH3’s interaction with PI4P may also affect glycosylation, an essential event for cell biology and known to drive the progression of different types of cancer. These topics should be addressed in more detail, as it would facilitate the reading.

- It has been demonstreted elsewhere that GOLPH3 modulates the activity of GNT-V by affecting Golgi apparatus function and regulating the glycosylation processes that are crucial for tumor progression. This interaction helps drive cancer cell growth, survival, and metastasis through altered glycosylation patterns. Understanding how GOLPH3 modulates GNT-V could lead to novel therapeutic strategies targeting glycosylation pathways in cancer. Since changes in the expression and activity of GNT-V are considered a hallmark in cancer, I suggest the authors describe how GOLPH3 modulates GNT-V.

- α-Dystroglycan is associated with a genetically-driven disease. Is there any relationship with the expression of the GOLPH3 protein? Please, explain it in more detail.

- Please discuss in more depth how GOLPH3 regulates cytokinesis and contributes to the uncontrolled cell division observed in cancer, and whether targeting this function could offer a new therapeutic approach.

- At the end of the manuscript, the authors briefly describe the relationship between GOLPH3 and the acquisition of drug resistance in cancer, which is currently an important obstacle faced by oncologists. The authors could discuss this highly relevant topic in oncobiology in more detail?

Comments on the Quality of English Language

The manuscript should be reviewed by a native English speaker.

Author Response

The reviewers’ comments are in Italics. Our responses are non-italicized

-Regarding the impact of GOLPH3 on tumor biology, I suggest the authors describe in more detail the role of the protein, which is known to be located in the Golgi apparatus, in events such as described below.

- GOLPH3 and its interaction with Phosphatidylinositol 4-phosphate (PI4P) is an essential phenomenon for maintaining the structure and function of the Golgi apparatus. This interaction modulates the activation of the mTOR pathway, which is central to regulating cell growth, metabolism, and survival. GOLPH3’s interaction with PI4P may also affect glycosylation, an essential event for cell biology and known to drive the progression of different types of cancer. These topics should be addressed in more detail, as it would facilitate the reading.

To meet the reviewer’s request, we changed the text of the manuscript on pages 2,3 and 8.

Specifically we changed the title of section 2 to “ Role of GOLPH3 family proteins in Golgi structure maintenance and glycosylation”. We amended section 2 to clarify that PI(4)P and its interacting protein GOLPH3 play an essential role for maintaining Golgi structure and function (Page 2, Lines 76-96). PI(4)P-GOLPH3 also affects glycosylation, an essential cellular process which has a great impact on cancer signaling, tumor progression and metastasis (Page 3, Lines 140-142).

In addition, we amended the text on page 8, Lines 297-299 as follows:

“Compelling evidence indicates that GOLPH3 and its interaction with PI(4)P, modulates the mTOR signaling pathway, which controls cellular growth, metabolism, and survival”.

- It has been demonstreted elsewhere that GOLPH3 modulates the activity of GNT-V by affecting Golgi apparatus function and regulating the glycosylation processes that are crucial for tumor progression. This interaction helps drive cancer cell growth, survival, and metastasis through altered glycosylation patterns. Understanding how GOLPH3 modulates GNT-V could lead to novel therapeutic strategies targeting glycosylation pathways in cancer.

A recent work has shown that GOLPH3 binds a large array of Golgi resident enzymes that act in several glycosylation pathways including N-linked glycosylation, O-mannosylation and proteoglycan synthesis (ref. 45 of the manuscript), but it fails to bind GNT-V. To our knowledge there are no other studies investigating how GOLPH3 modulates GNT-V. Because GOLPH3 was not reported to bind GNT-V, we could not address this issue in the revised manuscript.

- α-Dystroglycan is associated with a genetically-driven disease. Is there any relationship with the expression of the GOLPH3 protein? Please, explain it in more detail.

GOLPH3 interacts with Protein O-Linked Mannose β-1,2-N-Acetlyglucosaminyltransferase 1 (POMGnT1), a glycosyltransferase involved in the O-mannosylation of α-dystroglycan (ref. 35). Importantly, O-mannosylation of α-dystroglycan reduces ligand binding leading to various forms of congenital muscular dystrophies (ref. 37). However, the role of GOLPH3 related congenital muscular dystrophies has not been reported.

See the text on page 3, Lines 140-142:

“Further investigation will clarify whether loss of GOLPH3 might be involved in the development of the muscle-eye-brain disease.”

- Please discuss in more depth how GOLPH3 regulates cytokinesis and contributes to the uncontrolled cell division observed in cancer, and whether targeting this function could offer a new therapeutic approach.

To meet the reviewer’s request, we amended the paragraph on pages 9 (Lines 380-385) and 10 (Lines 411-427) and the conclusions (page 11, Lines 444-447). We discussed in more depth how GOLPH3 regulates cytokinesis and how targeting cytokinesis failures could offer a new therapeutic approach to selectively kill polyploid cancer cells that are much more sensitive to cytokinesis failures.

- At the end of the manuscript, the authors briefly describe the relationship between GOLPH3 and the acquisition of drug resistance in cancer, which is currently an important obstacle faced by oncologists. The authors could discuss this highly relevant topic in oncobiology in more detail?

As requested, we discussed in more detail the relationship between GOLPH3 and the acquisition of drug resistance in cancer. See the changes on page 11, Lines 435-441.

Reviewer 2 Report

Comments and Suggestions for Authors

The authors of this manuscript argue that crosstalk between GOLPH3 and mTOR is actually a molecular driver in cancer development, a new oncogene. For this reason, they analyze the progress that has been made in understanding the molecular and cellular function of GOLPH3 in the development and maintenance of the status quo of malignant tissues. In this work, the role of this protein in the glycosylation process is described in great detail. Then, they indicate that GOLPH3 enhances mTOR signaling in many types of cancers and also postulate its role during cytokinesis, proposing a molecular mechanism of this process. In conclusion, they emphasize that GOLPH3, through its significant influence on mTOR signaling, cytokinesis and glycosylation, may constitute an important molecular target for novel anticancer therapy. The manuscript is very well written and illustrated (three excellent figures). And although it touches on complex molecular processes, it is at the same time easy to read. In my opinion this interesting work contains no errors, omissions or shortcomings.

Author Response

The reviewers’ comments are in Italics. Our responses are non-italicized

The authors of this manuscript argue that crosstalk between GOLPH3 and mTOR is actually a molecular driver in cancer development, a new oncogene. For this reason, they analyze the progress that has been made in understanding the molecular and cellular function of GOLPH3 in the development and maintenance of the status quo of malignant tissues. In this work, the role of this protein in the glycosylation process is described in great detail. Then, they indicate that GOLPH3 enhances mTOR signaling in many types of cancers and also postulate its role during cytokinesis, proposing a molecular mechanism of this process. In conclusion, they emphasize that GOLPH3, through its significant influence on mTOR signaling, cytokinesis and glycosylation, may constitute an important molecular target for novel anticancer therapy. The manuscript is very well written and illustrated (three excellent figures). And although it touches on complex molecular processes, it is at the same time easy to read. In my opinion this interesting work contains no errors, omissions or shortcomings.

We thank the reviewer for their comments.

Reviewer 3 Report

Comments and Suggestions for Authors

Frappaolo et al. present a review manuscript about the role of GOLPH3, a Golgi-resident oncogene. They present an overview about GOLPH3 function in glycosylation, mTOR signaling and cytokinesis and what the links of these functions to tumorigenesis are. The manuscript is highly interesting, concise and very well written and could (almost) be published as is. Yet, I do have some suggestions for improvement that may be considered by the authors.

  1. Mostly throughout the manuscript the authors describe GOLPH3 as a trans-Golgi protein, but in lane 246 ff. they seem to insinuate that it localizes to the TGN, where most of the PI4P is. Also Scott et al. in their Nature paper (and maybe others) localize GOLPH3 to the TGN. Maybe the authors can clarify the localization, and/or, if controversial, point out that this is not fully established yet.

  1. Maybe a table with all known interaction partners could be added?

  1. Fig. 3. Why is there PIP2 in the membrane at the cleavage furrow, is it converted from PIP4? And why is there PI4P in the complex, shouldn´t that be in a membrane as well? Please explain that in the legend and/or text.

Author Response

The reviewers’ comments are in Italics. Our responses are non-italicized

  1. Mostly throughout the manuscript the authors describe GOLPH3 as a trans-Golgi protein, but in lane 246 ff. they seem to insinuate that it localizes to the TGN, where most of the PI4P is. Also, Scott et al. in their Nature paper (and maybe others) localize GOLPH3 to the TGN. Maybe the authors can clarify the localization, and/or, if controversial, point out that this is not fully established yet.

To meet the reviewer’s request, we clarified in the manuscript that GOLPH3 has been localized to the trans-Golgi as well as to the TGN (page 1, Lines 30,31).

  1. Maybe a table with all known interaction partners could be added?

To meet the reviewer’s request, we included a table (Supplementary Table S1) describing all the interaction partners that were mentioned in the review.

  1. Fig. 3. Why is there PIP2 in the membrane at the cleavage furrow, is it converted from PIP4? And why is there PI4P in the complex, shouldn´t that be in a membrane as well? Please explain that in the legend and/or text.

To meet the reviewer’s request, we amended Figure 3 and Figure 3 legend. Indeed PI(4)P is converted to PI(4,5)P2 in the cleavage furrow membrane.

See the changes in the new Figure 3 legend  and in the new Figure 3.

Round 2

Reviewer 1 Report

Comments and Suggestions for Authors

I thank the authors for their efforts to improve the quality of the manuscript